# Synthesis, Characterization and Biological Activities of Zinc Oxide Nanoparticles Derived from Secondary Metabolites of *Lentinula edodes*

**DOI:** 10.3390/molecules28083532

**Published:** 2023-04-17

**Authors:** Zeemal Seemab Amin, Muhammad Afzal, Jamshaid Ahmad, Naveed Ahmed, Basit Zeshan, Nik Haszroel Hysham Nik Hashim, Chan Yean Yean

**Affiliations:** 1Department of Basic and Applied Chemistry, Faculty of Science and Technology, University of Central Punjab, Avenue 1, Khayaban-e-Jinnah Road, Johar Town, Lahore 54590, Pakistan; 2Department of Medical Education, Sharif Medical and Dental College, Lahore 54000, Pakistan; 3Department of Medical Microbiology and Parasitology, School of Medical Sciences, Universiti Sains Malaysia, Kubang Kerian 16150, Malaysia; 4Faculty of Sustainable Agriculture, Universiti Malaysia Sabah (UMS), Sandakan 90509, Malaysia

**Keywords:** mycosynthesis, nanomaterial, bioactive compounds, in vitro activities, in vivo activities

## Abstract

Zinc oxide nanoparticles (ZnO NPs) are the second most prevalent metal oxide, owing to their characteristics of low cost, safe, and easily prepared. ZnO NPs have been found to exhibit unique properties which show their potential to be used in various therapies. Numerous techniques have been devised for the manufacture of zinc oxide because it is one of the nanomaterials that has received major research interest. Mushroom sources are proven to be efficient, ecologically friendly, inexpensive, and safe for humankind. In the current study, an aqueous fraction of methanolic extract of *Lentinula edodes* (*L. edoes*) was used to synthesize ZnO NPs. The biosynthesis of ZnO NPs was achieved by using the reducing and capping capability of an *L. edodes* aqueous fraction. Bioactive compounds from mushroom, such as flavonoids and polyphenolic compounds, are used in the green synthesis process to biologically reduce metal ions or metal oxides to metal NPs. Biogenically synthesized ZnO NPs were further characterized by using UV–Vis, FTIR, HPLC, XRD, SEM, EDX, zeta sizer and zeta potential analyses. The FTIR showed the functional group at the spectra in the range 3550–3200 cm^−1^ indicated the presence of the hydroxyl (OH) group, while bands in the range 1720–1706 cm^−1^ indicated C=O carboxylic stretches bonds. Furthermore, the XRD pattern of ZnO NPs created in the current study was found to be nanocrystals which are hexagonal. The SEM analysis of ZnO NPs showed spherical shapes and size distributions in the range 90–148 nm. Biologically synthesized ZnO NPs have substantial biological activities including antioxidant, antimicrobial, antipyretic, antidiabetic and anti-inflammatory potential. Biological activities showed significant antioxidant (65.7 ± 1.09), antidiabetic (85.18 ± 0.48), and anti-inflammatory potential (86.45 ± 0.60) at 300 µg inhibition in paw inflammation of (1.1 ± 0.06) and yeast-induced pyrexia (97.4 ± 0.51) at 10 mg in a dose-dependent manner. The outcomes of this research indicated that ZnO NPs significantly reduced inflammation and have the ability to scavenge free radicals and prevent protein denaturation, while also indicating their possible use in food and nutraceutical applications to treat various ailments.

## 1. Introduction

Nanotechnology is the most significant area of dynamic research in the scientific era. In recent times, nanotechnology has attained worldwide attention as a dynamically developing area of scientific interest [1]. Nanotechnology is concerned with developing and using particles up to 200 nm in size [2]. Based on specific factors including size, dispersion and morphology, the research on nanoparticles is altogether new and growing more advanced [3]. It has the potential to revolutionize many different industries by manipulating matter at the nanoscale. It enables scientists to develop extremely small, yet high-performance devices, materials and systems that can be used in a variety of applications, such as drug delivery, biosensors, energy storage, mechanics, optics, single-electron transistors, space industries, non-linear optical devices and photo-electrochemical applications [4]. Different techniques (chemical, physical and biological) can be used to create nanoparticles, which have a wide range of properties and applications.

Mushroom-based production of zinc oxide nanoparticles (ZnO NPs) has been described previously, but the literature on their various biological activities is presently lacking [5]. In addition, ZnO NPs exhibit remarkable capability in biological applications such as biological labeling, gene delivery, biological sensing, drug delivery and nano-medicine [6,7]. Mushroom-based synthesis of nanoparticles is an innovative method that has an extensive range of uses in the food industry, in agriculture and in medicine. Due to their toxicity, conventionally produced nanomaterials are rarely used in the therapeutic domain. Because of the physio-chemical characteristics of nanoparticles from mushrooms, this approach has the added benefit of NPs having a longer lifespan, which helps to overcome the drawbacks of traditional chemical and physical methods in the synthesis of NPs [8,9].

Green synthesis of metallic NPs can integrate a wide range of biological materials (e.g., bacteria, fungi, algae and plant extracts). This approach offers a sustainable, cost-effective alternative to traditional synthetic routes, with potential applications in the development of synthetic materials and catalysts. In many fungi, extracellular metabolites were observed that were capable of complex lysis and ligand-mediated dissolution. Several known metabolites exhibit these properties, including amino acids, carboxylic acids, phenolic compounds and siderophores. Among the strong chelating agents produced by fungi, oxalic acid is one of the most abundant. Oxalic acid’s extraction efficiency decreases with decreasing pH and increases with increasing pH. Therefore, Zn can be dissolved by oxalic acids and protonation in soil microenvironments close to fungi through oxalic acids and protonation. As a result, Zn is locally distributed and then reproduced as Zn oxalate elsewhere [10].

A variety of biomolecular synthesis methodologies determine variables such as solvents, temperatures, pressures and pH levels, which affect the biosynthesis of the NPs [11]. There have been attempts to commercialize the cultivation of mushrooms for human benefit since they are abundant sources of a range of secondary metabolites that can be used for medical and therapeutic purposes [12,13]. Shiitake mushroom (*Lentinula edodes)*, is the most commercially significant mushroom developed on wood. It is considered an essential ingredient for many Chinese and Japanese dishes [14]. Due to favorable climatic conditions, *L. edodes* cultivation is growing in Brazil using eucalypt logs, especially in the South and Southeastern regions [15]. *L. edodes* is ideal among functional mushrooms for the widespread research of its bioactivity, primarily the isolation of pure compounds which have important pharmaceutical use.

Bioactive compounds derived from *L. edodes* have been linked to medicinal potentials such as anti-tumor, anti-carcinogenic, anti-viral, preventative blood pressure increase in cases of hypertension, and hypocholesterolemic properties. Antifungal, antibacterial, and antiviral activities have also been described [16,17]. Due to the abundance of these phytochemicals, *L. edodes* have a greater potential for the efficient production of NPs, which can be employed to treat clinical ailments [18]. The present study aimed to achieve the green synthesis of ZnO nanoparticles from an aqueous fraction of the methanolic extract of *L. edodes*. The purpose of this study is to synthesize ZnO NPs from *L. edodes* using green synthesis methods. In comparison with synthetic techniques, mushroom extracts are a simple and inexpensive way to make ZnO NPs and can provide well-characterized nanoparticles. An integration of different methods of isolation, purification and optimization of NPs is presented in this study. The potency of ZnO NPs as free radical scavengers and antidiabetic agents, as well as access to more biological activity (anti-inflammatory, hemolytic and antibacterial), has been discussed as well.

## 2. Results

### 2.1. HPLC Analysis

The quantitative analysis of an aqueous fraction of *L. edodes* indicates the presence of phenolics and flavonoids. Among those, quercetin was the major component (45 mg/g), followed by benzoic acid (13.239 mg/g), sinapic acid (9.183 mg/g) and a very low concentration of m-coumaric acid (1.712 mg/g) as shown in Table 1 (Appendix A).

### 2.2. Characterization of Nanoparticles

#### 2.2.1. UV-Visible Spectroscopy

The zinc oxide nanoparticles (ZnO NPs) were dispersed in an aqueous fraction of methanolic extract with a concentration of 1:1, and then the solution was used to check the UV-Vis analysis. The physical analysis confirmed the production of ZnO NPs. As a result of the reaction, the reaction mixture’s color shifted from yellow to light yellow to milky white, indicating the synthesis of ZnO NPs. The figure below (Figure 1) illustrates the UV absorption spectrum of mushroom aqueous fraction and ZnO nanoparticles. As a result of the bioactive compounds present in mushroom aqueous fraction, a peak was observed at 281 nm. There was a peak at 363.3 nm in the spectrum of *L. edodes* ZnO NPs, indicating intrinsic bandgap absorption, which confirms that the nanoparticles were synthesized (Appendix A).

#### 2.2.2. FTIR Analysis

Using a Perkin-Elmer spectrometer system, a Fourier transform infrared spectroscopy (FT-IR) analysis of ZnO NPs was carried out to maintain the results of the HPLC. A FTIR spectroscopic approach was used to investigate possible phytochemical functional groups responsible for the reduction of zinc nitrate to ZnO NPs in the *L. edodes.* With a peak range of 300–4000 cm^−1^, the objective was to locate distinctive peaks and functional groups at a resolution of 4 cm^−1^. The bands at 3550–3200 cm^−1^ indicated the presence of the hydroxyl (OH) group, while bands at 1720–1706 cm^−1^ indicated the presence of carbonyls (C=O) of carboxylic acid (Appendix A and Figure 2).

#### 2.2.3. XRD Analysis

The XRD analysis of prepared ZnO NPs is shown in Figure 3. The product’s well-crystalline particle structure is hexagonal wurtzite, as shown by the product’s narrow and strong diffraction peaks. The peak values are 31.72, 34.28, 36.2, 45.68, 56.64, 62.68 and 69, corresponding to (80.17), (68.88), (97.22), (42.68), (68.11), (63.68) and (63.2), respectively, for the reflection lines of hexagonal wurtzite ZnO (JCPDS36-1451) (Figure 3). By comparison with the card data, all of the diffraction peaks were in the hexagonal zinc oxide phase. The product’s well-crystalline particle structure is shown by the product’s narrow and strong diffraction peaks.

#### 2.2.4. SEM Analysis

Individual ZnO NPs and various clusters were seen in the SEM image. According to Figure 4, the majority of elements are spherical in shape and aggregate into bigger particles with an uncertain geometry. ZnO NPs’ SEM and size distribution demonstrate a small range of particle sizes with a diameter of about 200 nm. In Figure 4, (a) shows the ZnO NPs, and (b) shows the size distribution of NPs. By using ImageJ^®^ software and SEM images obtained from NPs preparations, the particle diameter was calculated. For each preparation, particles with a diameter of 148.1 nm were measured. With the help of the origin software, version 2022 (OriginLab Corporation, Northampton, MA, USA), the frequency distribution of particle size was shown. The x-axis displays the particle diameter in micrometers, and the y-axis indicates the proportion of particles at a particular diameter (Appendix A).

#### 2.2.5. EDX Analysis

The EDX analysis confirms the presence of zinc oxide nanoparticles grown by the biosynthesized method. The elemental analysis of the ZnO NPs yielded ~2.70% of zinc and ~26.17% of oxygen, which proves that the produced ZnO NP is in its purified form (Figure 5).

#### 2.2.6. Zeta Potential

The value of the zeta potential (−30 mV to +30 mV) suggests the potential stability of the colloidal system. The produced NPs are shown in Figure 6 with a Zeta potential value of −16.2 mV, which clearly shows that they are relatively stable.

#### 2.2.7. Size Distribution Analysis (Zeta Sizer)

The size of the nanoparticles is 144.2 nm, as shown in Figure 7. ZnO NPs colloidal solution produced from 100 mM zinc nitrate solution and culture supernatant added in a 1:1 ratio produced the best results.

### 2.3. Biological Activities

#### 2.3.1. Antioxidant Potential

Zinc oxide nanoparticles of aqueous fraction were examined for antioxidant potential at different concentration (50, 100, 150, 200, 250 and 300 µg). The results showed that the scavenging activity of nanoparticles on antioxidants was concentration dependent, such that by increasing the concentration, the percentage inhibition gradually increased (Table 2). The results showed that at 300 µg, ZnO nanoparticles give the maximum scavenging activity (65.7 ± 1.09) as compared to the standard of 91.02 ± 0.54 µg/mL, with IC50 values of 29.48 µg as compared to the standard of 5.477 µg/mL (Appendix A).

#### 2.3.2. Antidiabetic Activity

The results showed that aqueous fractions of NPs contained active phytochemicals checked by various spectroscopic examinations and contained antidiabetic assays. To check α-amylase inhibition nanoparticles of *L. edodes* aqueous fractions of methanolic extract, alpha amylase inhibition assay was performed at different concentrations. The results showed that with a decreased concentration, the percentage inhibition was decreased. The maximum % inhibition was observed at 300 µg/mL (85.18 ± 0.48), nearest to standard metformin (87.69 ± 1.57) at the same concentration, with IC50 values of 15.10 µg/mL, respectively, as compared to the standard of 12.48 µg/mL (Table 3).

#### 2.3.3. Hemolytic Activity

ZnO NPs of *L. edodes* aqueous fractions of methanolic extract did not show marked hemolysis. Titron shows 100% hemolysis at 0.1% used as a positive control (96.60 ± 0.10) and PBS as a negative control (2.31 ± 0.002). The data shown in Table 4 represent the hemolytic potential of ZnO NPs of *L. edodes* aqueous fractions at different concentrations. The highest hemolytic percentage (6.90 ± 0.06) was observed at 300 µg/mL, while the lowest value (2.19 ± 0.34) was noticed at 50 µg/mL. Overall, each concentration has less than 10% hemolysis (shown in Table 4), so NPs are non-hazardous to humans and thus safe.

#### 2.3.4. Antibacterial Activity

The chosen bacterial species includes Gram-positive (*S. aureus* (N315)) and Gram-negative bacteria (*E. coli* (CP034953) and *K. pneumoniae* (AJHE00000000)). The prepared ZnO NPs by the green method show a higher toxic effect against the bacteria *Klebsiella* and *S. aureus* and *E. coli*. A distinct inhibitory zone was visible for different ZnO NPs concentrations in Table 5 and Table 6. Appendix A demonstrated that the inhibitory zone grew when ZnO NPs concentrations were increased (Appendix A).

#### 2.3.5. In Vitro Anti-Inflammatory Activity

The current study showed in vitro anti-inflammatory ability of ZnO NPs of an aqueous fraction of *L. edodes* on denaturation of proteins inhibition, as shown in Figure 7. The results showed that ZnO NPs had excellent anti-inflammatory properties, which is concentration dependent, whereby as the concentration increases, the percentage inhibition also increases. The maximum % inhibition of diclofenac was observed at 300 µg/mL (89.76 ± 0.86) followed by 50 µg /mL (54.76 ± 0.57). Similarly, to diclofenac, ZnO NPs showed a similar trend of inhibition. The maximum inhibition was observed at 300 µg/mL (86.45 ± 0.60), nearest to the positive control, with an IC50 value of 6.74 µg/mL, as compared to the standard 4.97 µg/mL (Appendix A).

#### 2.3.6. In Vivo Anti-Inflammatory Activity

Rat paw edema was induced by using 0.1% carrageenan. The average size of the paw and percentage inhibition of ZnO NPs was measured (Table 7). The results showed a gradual increase in the percentage inhibition in paw edema at 4 h of carrageenan injection. By comparing each group under observation to the control group, a gradual increase in paw volume was found. The mushroom-based synthesized nanoparticles showed peak edema suppression within 4 h of carrageenan injection in a dose-dependent manner. Nanoparticles showed significant anti-inflammatory action (*p* < 0.01), similarly to diclofenac sodium, within 60 min of carrageenan injection (Appendix A).

#### 2.3.7. In Vivo Antipyretic Activity

The body temperature was lowered by ZnO NPs of an *L. edodes* aqueous fraction up to 4 h after its administration. The subcutaneous injection of yeast caused a significant increase in rectal temperature, which peaked at 101 F°. After treatment with mushroom-synthesized NPs at different concentrations (6, 8, 10 mg/kg), the temperature dropped to 99.1 F° an hour later, and it continued to drop for the next four hours, showing a significant drop in temperature that was almost identical to that caused by the positive control. Up to 2 h, nanoparticles at 6 mg/kg showed attenuated fever; this effect was not statistically (*p* > 0.05) significant and indicated a marginally significant (*p* < 0.05) change after 3 h (*p* < 0.01) following 4 h of treatment in comparison to the control. Management with paracetamol at a dose rate of 10 mg/kg significantly (*p* < 0.001) decreased (Table 8) (Appendix A).

## 3. Materials and Methods

### 3.1. High-Performance Liquid Chromatography (HPLC)

An HPLC analysis of mushroom aqueous fraction was used to quantify phenolics and flavonoids. Shim-packed CLC ODS C-18 (5 mm diameter, 2.5 cm height; 4.6 mm diameter, 2.5 cm height) columns were used. Using an aqueous fraction consisting of 10 mg/mL, the extracts were prepared. An amount of 20 mL mushroom extract was mixed with 94:6 hydrogen acetoacetate mobile phase A and with pH 2.27 mobile phase B (ACN100%), with the different parameters of 15% to 45% B in the first 15–30 min and 100% to 45% in the last 35–40 min. The 280 nm wavelength was used to measure all UV-visible detector spectra [19].

### 3.2. Synthesis of ZnO NPs

*Lentinula edodes* whole mushroom was washed and dried at room temperature (25 °C). The whole dried mushroom was crushed, powdered and soaked with methanol at room temperature. The filtrate was evaporated in the rotatory evaporator to yield the semi-solid extract. The extract was then further fractionated with n-hexane and distilled water. Biogenic production of ZnO NPs was carried out following the method by Selim et al. [20,21] with slight modifications. *L. edodes* aqueous fractions of methanolic extract (about 50 mL) were mixed with 100 mM of zinc nitrate hexahydrate (Zn(NO_3_)_2_·6H_2_O) (about 50 mL) and heated (65–85 °C) on a magnetic stirrer. When the extracted temperature reached 60 °C, the pH was adjusted to 12 by gently adding 0.1% NaOH drop by drop in a molar ratio of 1:1 while vigorously stirring, and the mixture was allowed to stand for about 2 h until a white precipitate formed. A cream paste was then formed by baking this mixture for another day at 70 °C in a hot air oven. A solution of ethanol and distilled water was used to extract and wash the paste repeatedly. After that, the paste was placed in a ceramic crucible cup and cooked for two hours at 400 °C in a furnace. The resulting white powder was kept for characterization in an airtight container [22].

### 3.3. Optimisation of Nanoparticles

UV-Vis spectroscopy was used to check the impact of pH on the formation of nanoparticles (Figure 5). No absorption peaks were observed at pH of 4 or 6. The absorption peaks were observed from 360 to 380 nm at pH of 10 and 12. At 363.4 nm, pH 12 showed a sharp peak, indicating that all of the zinc nitrate hexahydrates had been converted to ZnO NPs.

### 3.4. Characterization of ZnO Nanoparticles

Analysis of the UV-visible spectrum was used to track the emergence of ZnO NPs. A double-beam V-630 spectrophotometer operating in the 100–700 nm wavelength range was used for the analysis (Shimadzu UV-2500PC Series, Kyoto, Japan). Characterization of potent bioactive groups linked to the surface of nanoparticles was performed by using a Perkin-Elmer spectrometer system, with a resolution of 4 cm^−1^ within a peak range of 450–4000 cm^−1^ [23].

SEM was used to examine the morphology of ZnO NPs by (EVO LS10, Zeiss, Germany), which were prepared using an aqueous fraction of methanolic extract at an accelerating voltage of 12 keV. Merely by putting a slight sample on a carbon-coated copper, a grid sample was generated. SEM images were taken at various magnifications following a five-minute drying period under a mercury lamp. For elemental examination, the dried powdered ZnO NPs sample was drop-coated onto a carbon sheet. After that, analysis was performed using an EDX detector attached to the SEM [24].

An XRD analysis was performed using a powder X-ray diffractometer from STOE (Darmstadt, Germany) (operating voltage 40 kV, current 40 mA). An appropriate source was Cu Kα radiations (λ = 1.54 A°) with 2 θ range 20–80° for ZnO NPs and 10–80° for aqueous fraction of nanoparticles [25]. Debye Scherrer’s formula was used to calculate the crystallite size of ZnO NPs
$$\mathrm{Crystallite}\;\mathrm{Size}=\frac{0.9\lambda }{!\beta \;\cos\theta }$$
where λ = wavelength used for analysis 1.54 A t, β = Full width half maximum, and θ = scan range for analysis [26].

### 3.5. Biogical Activities

#### 3.5.1. Antioxidant Potential (DPPH Assay)

For estimation of antioxidants, a sample of 100 µL of NPs at different concentrations (50, 100, 150, 200, 250 and 300 µg) was mixed with 50 µL of DPPH in a 96-microtiter plate. This mixture was then incubated at ambient temperature for 30 min in the dark. Elisa reader microplates were used to detect absorbance at 630 nm. Ascorbic acid (Vit C) was used as standard at different concentrations in 1 mL of distilled water. The free radical scavenging activity was calculated by percentage inhibition using formula. Results were interpreted in terms of IC50 as sample was analyzed at different concentrations, and the IC50 value was calculated by drawing calibration curve through MS Excel Sheet [27].
$$\mathrm{DPPH}\;\mathrm{Percent}\;\mathrm{Inhibition}\;=\left(\frac{\mathrm{Absorbance}\;\mathrm{of}\;\mathrm{blank}\;-\;\mathrm{Absorbance}\;\mathrm{of}\;\mathrm{the}\;\mathrm{sample}}{\mathrm{Absorbance}\;\mathrm{of}\;\mathrm{blank}}\right)\times 100$$

#### 3.5.2. Antidiabetic Activity

The starch iodine method was performed to check α-amylase inhibition of ZnO NPs at various concentrations (50, 100, 150, 200, 250 and 300 µg). Initially, 5 µL of the α-amylase solution was incubated with 195 µL of NPs sample, which was prepared in DMSO and the standard of different concentrations for 10 min at 37 °C. Starch solution (50 µL) was added to the solution after the initial incubation, and it was then incubated for a further 60 min. Furthermore, 100 µL of a 1% iodine solution was added, and the Biotek ELISA plate reader (Winusky, VT, USA) was used to measure the absorbance at 630 nm [28].

Metformin was taken as positive control and PBS as a negative control. Percentage inhibition was measured by the following formula.
$$\%\;\mathrm{inhibition}\;=\left(\frac{\mathrm{Absorbance}\;\mathrm{of}\;\mathrm{control}\;-\;\mathrm{Absorbance}\;\mathrm{of}\;\mathrm{the}\;\mathrm{sample}}{\mathrm{Absorbance}\;\mathrm{of}\;\mathrm{control}}\right)\times 100$$

#### 3.5.3. Hemolytic Activity

The hemolytic activity of ZnO NPs of aqueous fraction at different concentrations (100, 150, 200, 250, 300 and 350 µg) of *L. edodes* was assessed to determine whether or not ZnO NPs have any cytotoxic effect on cells. Hemolytic activity was checked by some modifications of Powell and Shahzad et al. Absorbance were measured at 630 nm on a Biotek ELISA reader. The studies were performed three times, and mean absorbance values were noted. PBS was taken as the negative control and 0.1% Titron × 100 as the positive control. Percentage hemolysis was calculated by the following formula [29].
$$\%\;\mathrm{hemolysis}\;=\left(\frac{\mathrm{Absorbance}\;\mathrm{of}\;\mathrm{sample}\;-\;\mathrm{Absorbance}\;\mathrm{of}\;\mathrm{negative}\;\mathrm{control}}{\mathrm{The}\;\mathrm{absorbance}\;\mathrm{of}\;\mathrm{positive}\;\mathrm{control}}\right)\times 100$$

#### 3.5.4. Anti-Bacterial Activity

The disc diffusion technique was used to determine the antibiotic activity of the nanomaterials against a variety of bacterial species (Gram-positive (*Staphylococcus. aureus*) and Gram-negative bacteria (*E. coli*) and *Klebsiella pneumoniae*). The test was performed on nutrient agar. On agar plates, culture was spread using a glass spatula. A sample was applied to the disc at various amounts using a clean disc and incubated for 24 h at 37 °C. The Ampicillin disks were used as a positive control, with the DMSO as a negative control. After the incubation period, the zones of inhibition were measured using a clear reader [30].

#### 3.5.5. In Vitro Anti-Inflammatory Activity

The anti-inflammatory activity of the *L. edodes*-synthesized NPs of aqueous fraction was determined using a modified version of the BSA assay reported by Rajakumar et al. The turbidity of the sample was measured at 630 nm in an ELISA reader. The tests were conducted in triplicate, and the mean absorbance values were recorded. Diclofenac in water is used as standard. Results were interpreted in terms of IC50 values. Percentage inhibition was calculated by the formula given in Section 3.5.2 [31].

#### 3.5.6. In Vivo Anti-Inflammatory Activity

To perform in vivo anti-inflammatory activity, all animals were put in polypropylene cages with three per cage, kept at an ambient temperature of 25 °C and relative humidity of 55.65%, and kept on a 12 h light/dark cycle. Rats in all groups except the control received an injection of 0.1 mL of 1% carrageenan into their right hind paw. With the aid of a digital Vernier caliper, the size of rats’ paws was measured. The rats were given various ZnO NPs doses (200, 300, 400 mg/kg) and standard Diclofenac 15 mg/kg, i.p, half an hour before. Before carrageenan injection, the thickness of the paws was experimentally verified, at “0 h”, and subsequently at 1, 2, 3 and 4 h. The percentage inhibition of paw edema was calculated using the following formula [32].

The percentage (%) of inhibition of edema was calculated by:$$\%\;\mathrm{inhibition}\;=\frac{\mathrm{To}\;-\;\mathrm{Tt}}{\mathrm{To}}\times 100$$

#### 3.5.7. Antipyretic Activity

Yeast-induced pyrexia test was assessed to check the antipyretic potential of ZnO NPs. The mice were randomized into three groups. The body temperature was measured using a digital thermometer. Pyrexia was induced by subcutaneous injection of attenuated yeast (20% *w*/*v*) in normal saline. The precise 17th hour following the subcutaneous (s/c) shot was used to determine the body temperature. Group 2 was treated as a positive control (standard), receiving treatment with yeast and 100 mg/kg of paracetamol, while group 1 functioned as a normal control group, receiving just normal saline at a dose of 10 mL/kg [33].

### 3.6. Statistical Analysis

The results are shown as Mean SD in the tables. A two-way ANOVA was run using the software Graph Pad Prism, version 2022. The graphs’ values are all shown as mean SE.

## 4. Discussion

The biological synthesis of nanoparticles is an evolving method for diagnosis and treatment, with the benefits of using harmless bioactive molecules found in mushroom aqueous fraction, being economical and involving single-step approaches for their association, thus fascinating the researchers in [34]. The ZnO NPs were synthesized using an aqueous fraction of *L. edodes* methanolic extract containing stabilizing agents such as flavonoids and phenolics as the main bioactive compounds. The HPLC analysis of an aqueous fraction of methanolic extract indicates the presence of phenolic compounds in appropriate amounts, such as quercetin, benzoic acid, m-coumaric acid and sinapic acid, which may be responsible for the bio-reduction of the metal salts into ZnO particles. It is known that flavonoids and phenolic acids are both powerful hydrogen donors [35], which is responsible for a variety of biological activities because of their functional (hydroxyl and carboxyl) groups. Bioactive compounds identified from the HPLC analysis might be capped with ZnO NPs due to the reducing capability of flavonoids and phenolics [36].

The most useful approach for the structural characterization of NPs is UV-Vis spectroscopy. *L. edodes*-synthesized NPs exhibits a solid UV absorption band with an absorption peak at 363.3 nm (Figure 1) due to its surface plasmon resonance. In addition, these sharp peaks show that the particles are in nano-size, and the particle size distribution is narrow. This is in good agreement with previous studies which are in range of 360–370 nm [37,38]. From 20 to 40 Mm salt concentration, no absorption peaks were recorded. The greatest absorption peak was seen at 363 nm for 100 mM concentration, and the absorption peak for 50 and 70 mM concentrations was found at 360 nm. This band was found to be the result of the ZnO NPs valance electrons being excited. Similar findings have been documented by studies which described that the absorbance ranges of ZnO NPs were recorded in the range 310–370 nm at 80–100 mM salt concentration [31,39].

FTIR spectroscopy is frequently used to discover the biologically active molecules and some of the functional groups involved in producing ZnO NPs synthesized via the *L. edodes* [40]. It is a reliable method for substance identification and characterization since it provides an impression of the rotational and vibrational modes of the presentation of molecule [41]. As depicted in Figure 2, the FTIR spectra of ZnO NPs exhibit promising biomolecule functional groups information about the *L. edodes*-synthesized ZnO NPs identified at various wavenumber ranges. The peak range of 450–4000 cm^−1^ was used to identify potential biomolecules, and the bands at 3550–3200 cm^−1^ indicated the presence of the hydroxyl (OH) group. At the same time, bands at 1720–1706 cm^−1^ indicated the presence of carboxyl (C=O) groups (Figure 2), which confirms the role of –OH groups as reducing agents for the formation of NPs [42]. Zeta potential demonstrates stability that assures their stability. In the present study, the zeta potential analysis of nanoparticles was performed, which was approximately −16.2 mV, which indicates that biofunctionalized ZnO NPs from an aqueous fraction of *L. edodes* show greater stability (Figure 6). Dispersity and long-term stability are supported by the prevailing negative surface charge potential [43]. Zeta sizer indicated the size of the ZnO NPs, which is 144.2 nm, while an average size is 148 nm.

The X-ray diffraction technique (XRD) provides detailed information about the crystallographic structure, chemical composition, and physical properties of nanoparticles. Figure 3 depicts the prepared ZnO NPs XRD pattern peak positions for the hexagonal wurtzite ZnO (JCPDS36-1451) reflection lines, which are 31.72, 34.28, 36.2, 45.68, 56.64, 62.68 and 69, corresponding to (80.17), (68.88), (97.22), (42.68), (68.11), (63.68) and (63.2), respectively. The product’s well-crystalline particle structure is shown by the product’s narrow and strong diffraction peaks. A study depicted a comparison of the XRD pattern with the standard confirmed that the ZnO particles formed in the present experiments were nanocrystals with structure hexagonal wurtzite. Previous studies have found approximately similar findings for the crystallinity of nanoparticles [44].

The composition of the nanostructures found from mushroom aqueous fraction was studied using SEM. The sample is made up of electrons that combine with the atoms to generate signals that give details on the surface topography, composition and other features such as electrical conductivity [44]. ZnO NPs that have been synthesized can be seen in SEM images in a variety of different shapes, such as hexagonal and spherical nanoparticles. The aggregated ZnO NPs were generated with a particle size of less than 148.1 nm (Figure 4). Additionally, this study shows the eminent spherical and hexagonal structure, generally random and not constant, and it closely resembles those mentioned in the earlier literature [35].

Biologically synthesized ZnO NPs were checked for their ability to scavenge free radicals using the DPPH assay. Green synthesized ZnO NPs have almost the same antioxidant activity as mushroom aqueous fraction (Table 2). The maximum inhibition was 65.7 ± 1.09 at 300 µg for biologically synthesized ZnO NPs. The conversion of violet DPPH color to yellow demonstrated clearly the antioxidant impact of NPs. Ascorbic acid was used as the standard. These metabolites combine to form the ZnO NPs. They adsorbed on the surface of the ZnO NPs. Given their large surface-area-to-volume ratio, these ZnO NPs appear to have a strong propensity to interact with and decrease DPPH. It is similar to those described in the previous literature, which is in the range of 55–70% at different concentrations [31]. This study clearly shows that ZnO NPs synthesized by an *L. edodes* aqueous fraction of methanolic extract can act as a potential antioxidant. Smaller particle size and another factor together represent the phenomenon of the transfer of electron density from the oxygen atom to the odd electron located at the nitrogen atom in DPPH, which is attributed to ZnO NP antioxidant activity; this phenomenon indicates a decrease in the nanoparticles’ transition intensity at 630 nm. Antimicrobial activity of metal oxides ZnO, MgO and CaO powders were quantitatively assessed in culture media [45]. It was checked that identified active oxygen species produced by these metal oxide particles could be the main reason for their antibacterial activity (Table 5) [46]. The enzyme α-amylase, mostly present in saliva and pancreatic juice, is responsible for hydrolyzing -1,4-glucan polysaccharides such as starch. High blood glucose levels can be prevented by inhibiting this enzyme. The ZnO NPs showed reasonable α-amylase inhibition activity, and the results are presented in terms of IC50 values (Table 3). The IC50 values of the ZnO NPs (15.102 µg/mL) were higher than those of standard metformin (12.48 µ/mL), respectively, which confirmed that ZnO NPs showed better antidiabetic potential in terms of the a-amylase inhibition activity. Previous studies showed a similar trend [47,48]. ZnO nanoparticles showed excellent anti-inflammatory activity, which is concentration dependent, such that an increase in concentration means an increase in percentage inhibition. The maximum % inhibition of Diclofenac was observed at 300 µg/mL (89.76 ± 0.86) and then at 50 µg /mL (54.76 ± 0.57) with an IC50 value of 4.97 (Table 6). Similarly to Diclofenac sodium, ZnO nanoparticles showed a similar trend of inhibition. The maximum inhibition was observed at 300 µg/mL (86.45 ± 0.60). In vitro anti-inflammatory activity indicates that ZnO nanoparticles inhibited denaturation of BSA compared to previous studies of ZnO NPs which showed similar results [49], in which ZnO NPs were further assessed through in vivo bioactivities. In vivo anti-inflammatory activity was checked in albino mice through carrageenan-induced paw oedema method. The inhibition of inflammation was higher in the second and third hours but was lower in the first hour. Table 7 illustrates the larger-percentage inhibitory impact. The inhibition of NMs was greater than that of free drugs on a weight basis [50]. To check the antipyretic effect of ZnO NPs, the yeast-induced pyrexia method was used, and the animals were cured with 10% Brewer’s yeast per body weight. The selection of mice was based on body temperature. The mice selected for activity were those whose body temperature was 0.6 C higher after 15h of treatment with yeast. The mice were then treated with mushroom-synthesized ZnO NPs with different doses. The antipyretic effect of the NMs was greater than drugs, as shown in Table 8, with a previous study giving comparable results [50].

## 5. Conclusions

In the current study, the ZnO NPs were produced using the green method, which is a cost-effective, pollution-free, and eco-friendly method for antipyretic, antidiabetic, antioxidant, antimicrobial, and anti-inflammatory activity. ZnO NPs were stabilized in the present study by bioactive compounds. Both reducing and stabilizing agents can be obtained from an aqueous fraction of *L. edodes*. Various techniques such as UV-Vis, FTIR, XRD, SEM, zeta sizer and zeta potential analysis were performed to characterize the ZnO NPs. The XRD pattern verified the sample’s crystallinity and pristine phase. According to the findings, ZnO NPs with a prominent spherical and hexagonal shape were produced. The majority of the particles collected were cylindrical, and the particle size may be ordered by varying extract and salt amounts. The effect of compounds such as flavonoids, polyphenols and a carboxylic acid, in particular owing to the existence of carboxylic and hydroxyl groups that have reacted with the zinc nitrate, caused the NPs to develop. The characterized ZnO NPs can be used to reduce sugar levels, in inflammations, and as an antibiotic. Biosynthesized ZnO NPs produced from *L. edodes* will be crucial in pharmacological and biological areas such as medication transport and makeup. In the future, biogenic liposomal nanoparticles may prove effective in treating a variety of diseases due to their potent biological activity.

## Figures and Tables

**Figure 1 molecules-28-03532-f001:**
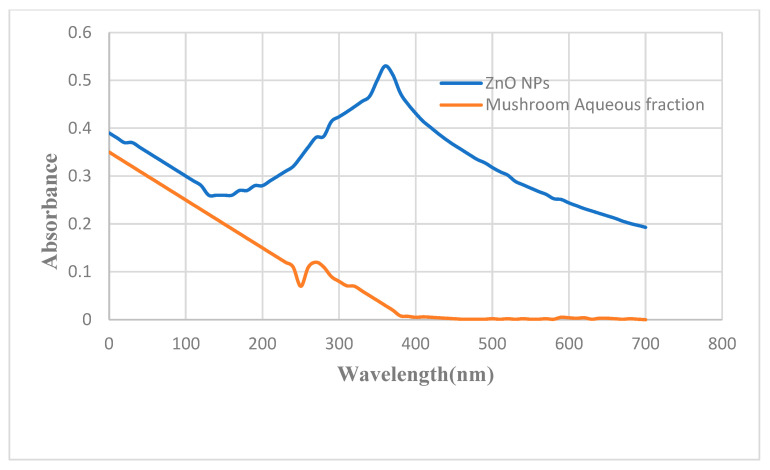
UV-Vis spectrum of ZnO NPs and *L. edodes* aqueous fraction.

**Figure 2 molecules-28-03532-f002:**
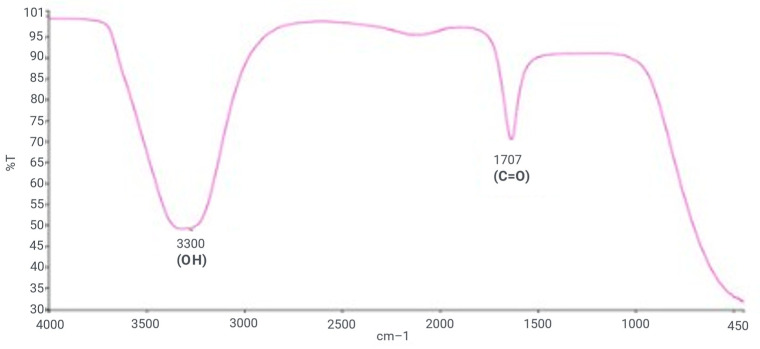
FT-IR Spectrum of ZnO NPs from aqueous fraction of *L. edodes*.

**Figure 3 molecules-28-03532-f003:**
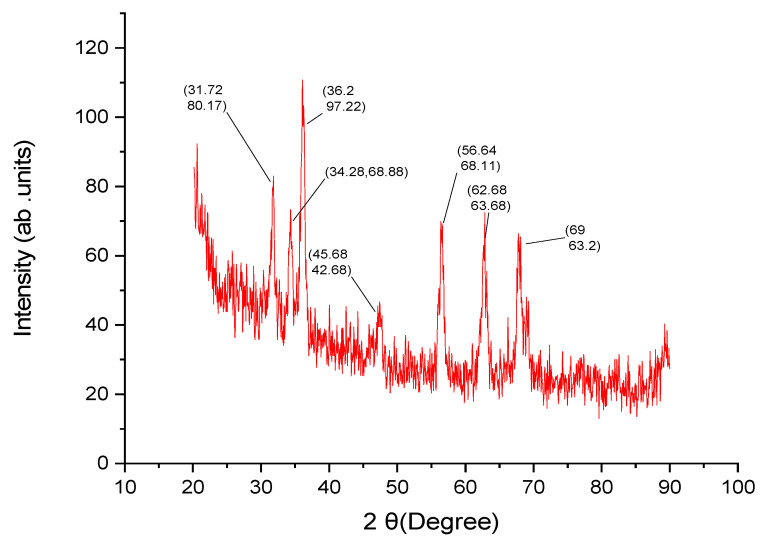
The XRD spectra of biosynthesized ZnO NPs from aqueous fraction of *L. edodes*.

**Figure 4 molecules-28-03532-f004:**
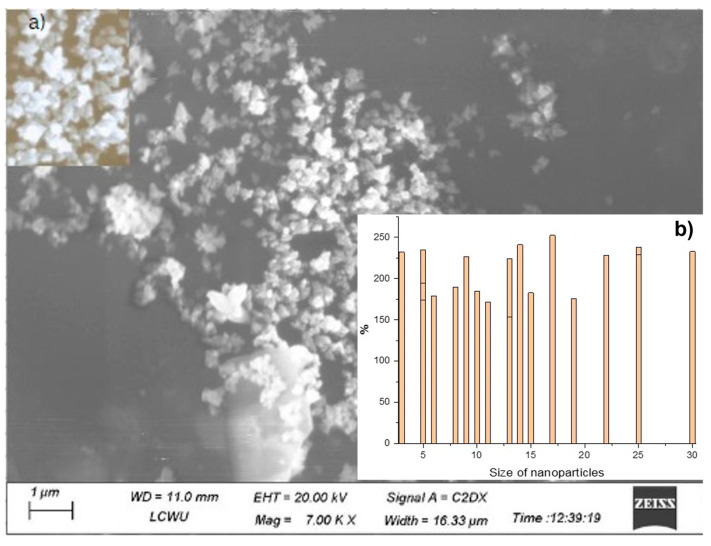
SEM images of ZnO NPs: (**a**) shows the ZnO NPs; (**b**) shows the size distribution of NPs.

**Figure 5 molecules-28-03532-f005:**
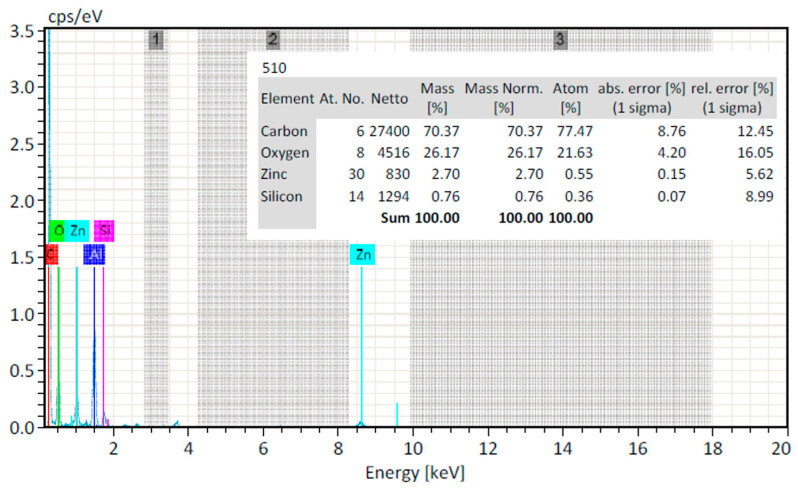
EDX analysis of ZnO NPs.

**Figure 6 molecules-28-03532-f006:**
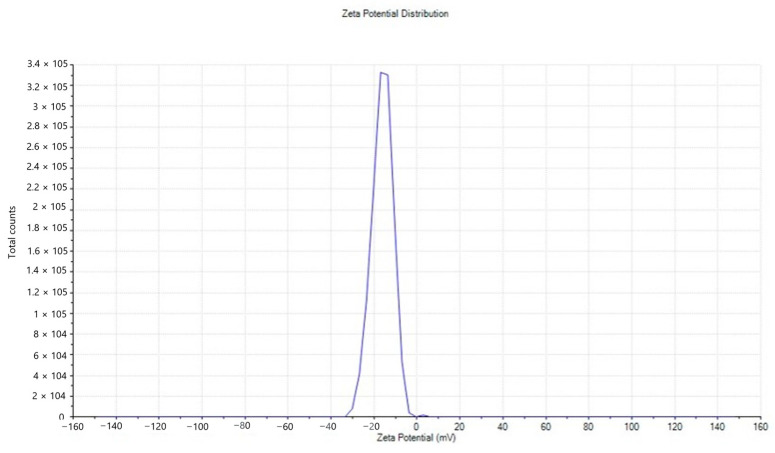
Zeta potential distribution of ZnO NPs.

**Figure 7 molecules-28-03532-f007:**
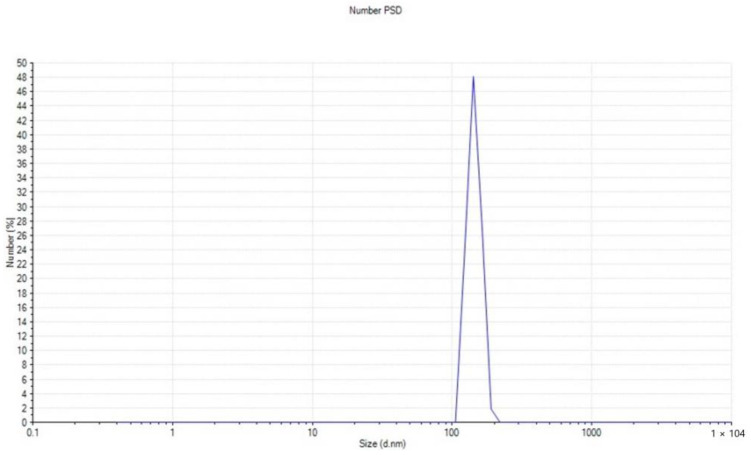
Particle size distribution of ZnO NPs from *Lentinula edodes*.

**Table 1 molecules-28-03532-t001:** HPLC screening of phenolics and flavonoids compounds (ppm) of aqueous fractions of methanolic extract of Lentinula edodes.

Sr. No	Name of Compounds	Retention Time (min)	Area (%)	Concentration (ppm)
1	Quercetin	3.521	2.53	45
2	Benzoic acid	14.847	3.3	13.239
3	m-coumaric acid	20.453	3.8	1.712
4	Sinapic acid	26.840	18.7	9.183

**Table 2 molecules-28-03532-t002:** Free radical scavenging activity of ZnO NPs of aqueous fraction of *L. edodes*.

Concentrations µg/mL	DPPH Inhibition (%) (Mean ± SD)	IC50	Ascorbic Acid (Mean ± SD)	IC50
50	28.78 ± 0.54	29.84	51.15 ± 1.73	5.477
100	35.60 ± 0.64		61.52 ± 0.23	
150	41.7 ± 0.68		69.03 ± 0.67	
200	48.6 ± 0.80		75.37 ± 0.08	
250	56.5 ± 0.61		85.11 ± 0.63	
300	65.7 ± 1.09		91.02 ± 0.54	

The values are expressed as the Mean ± SD.

**Table 3 molecules-28-03532-t003:** α-Amylase inhibition (%) of ZnO NPs of aqueous fraction of *L. edodes* methanolic extract. The values are presented as the Mean ± SD.

Concentrations µg/mL	α-Amylase Inhibition (%) (Mean ± SD)	IC50	Metformin Inhibition (%) (Mean ± SD)	IC50
50	44.99 ± 0.99	15.102	46.35 ± 0.96	12.48
100	50.72 ± 0.98		54.33 ± 0.43	
150	58.57 ± 0.90		61.4 ± 1.09	
200	67.76 ± 1.14		68.53 ± 0.59	
250	78.07 ± 0.90		77.92 ± 0.39	
300	85.18 ± 0.48		87.69 ± 1.57	

**Table 4 molecules-28-03532-t004:** The hemolytic potential of ZnO NPs of aqueous fraction of *L. edodes* methanolic extract.

Sr. #	Concentrations µg/mL	Percentage Hemolysis (%) (Mean ± SD)
1	50	2.19 ± 0.34
2	100	2.64 ± 0.98
3	150	3.52 ± 0.60
4	200	4.21 ± 1.14
5	250	4.74 ± 0.70
6	300	6.90 ± 0.06
7	0.1% Titron X	96.60 ± 0.10
8	PBS	2.31 ± 0.002

**Table 5 molecules-28-03532-t005:** ZnO NPs zone of inhibition in mm.

Sr. #	Microorganism	Zone of Inhibition in mm Concentration µg			
		A 500	B 750	C 1000	D 1500
1	*Staphylococcus aureus*	9	11	12	14
2	*Klebsiella pneumoniae*	8	10	13	15
3	*Escherichia coli*	8	9	9	10

**Table 6 molecules-28-03532-t006:** Percentage inhibition of protein denaturation of NPs of aqueous fraction of *L. edodes.* The values are expressed as the Mean ± SD.

Sr. #	Concentrations µg/mL	% Denaturation Proteins (Mean ± SD)	IC50	Dicloran Inhibition (%) (Mean ± SD)	IC50
1	50	50.04 ± 0.44	6.74	54.76 ± 0.57	4.97
2	100	58.17 ± 0.37		57.76 ± 2.01	
3	150	65.22 ± 0.81		65.76 ± 1.18	
4	200	72.69 ± 0.43		73.76 ± 0.94	
5	250	79.99 ± 0.43		81.76 ± 1.56	
6	300	86.45 ± 0.60		89.76 ± 0.86	

**Table 7 molecules-28-03532-t007:** In vivo anti-inflammatory effects of ZnO NPs of *L. edodes* aqueous fractions of methanolic extracts on carrageenan-induced inflammation.

Sample Fraction	Dose mg/kg Body Weight	Change in Paw Thickness (mm) 0 h	1 h	2 h	3 h	4 h
Nanoparticles	6	1.31 ± 0.05 18.75%	1.41 ± 0.04 25.39%	1.51 ± 0.03 31.36%	1.62 ± 0.03 42.14%	1.31 ± 0.03 53.73%
	8	1.21 ± 0.03 28.9%	1.3 ± 0.06 35%	1.34 ± 0.02 36.19%	1.51 ± 0.06 46.07%	1.1 ± 0.03 60.99%
	10	1.12 ± 0.04 34.91%	1.31 ± 0.02 37.61%	1.42 ± 0.05 40.83%	1.46 ± 0.04 47.85%	1.1 ± 0.06 63.21%
Neg Control	6	1.38 ± 0.04 30%	1.47 ± 0.03 1%	1.57 ± 0.04 2%	1.65 ± 0.04 15%	1.64 ± 0.03 0%
	8	1.41 ± 0.02 40%	1.52 ± 0.06 2%	1.63 ± 0.03 3%	1.72 ± 0.04 1%	1.73 ± 0.03 0%
	10	1.25 ± 0.05 3%	1.35 ± 0.03 1%	1.45 ± 0.04 3%	1.52 ± 0.03 1%	1.53 ± 0.06 0%
Diclofenac Sodium Standard	6	1.12 ± 0.03 31	1.31 ± 0.04 45%	1.41 ± 0.02 58%	1.53 ± 0.02 61%	1.2 ± 0.02 70%
	8	1.14 ± 0.02 35%	1.21 ± 0.06 47%	1.32 ± 0.03 56%	1.41 ± 0.04 67%	1.13 ± 0.03 75%
	10	1.25 ± 0.05 38%	1.35 ± 0.03 45%	1.45 ± 0.04 57%	1.56 ± 0.03 66%	1.21 ± 0.06 76%

**Table 8 molecules-28-03532-t008:** Antipyretic effects of ZnO nanoparticles on yeast-induced pyrexia in rat.

Drug	Dose mg/kg Body Weight	Temperature Before Yeast Injection (F°)	0 h	1 h	2 h	3 h	4 h
Nanoparticles	6	97.5	101.2 ± 0.37	99.7 ± 0.34	98.6 ± 0.02	97.6 ± 0.37	97.3 ± 0.5
	8	97.4	101 ± 0.31	99.5 ± 0.41	98.4 ± 0.7	97.3 ± 0.22	97.3 ± 0.4
	10	98.1	100.3 ± 0.41	99.3 ± 0.05	98.1 ± 0.3	97.3 ± 0.4	97.4 ± 0.51
Negative Control		97.6	100.1±0.51	100.2 ± 0.5	100.2 ± 0.51	100.2 ± 0.02	101.1 ± 0.4
Paracetamol Standard	6	97.6	100.1 ± 0.32	98.4 ± 0.4	97.9 ± 0.21	97.8 ± 0.03	97.7 ± 0.3
	8	98.6	100 ± 0.31	98.3 ± 0.3	97.5 ± 0.5	97.7 ± 0.04	97.6 ± 0.4
	10	98.4	100.1 ± 0.41	98.1 ± 0.3	97.5 ± 0.46	97.6 ± 0.02	97.6 ± 0.03

## Data Availability

More data related to this study can be accessed upon a reasonable request to the corresponding author or dr.afzal@ucp.edu.pk.

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
