# Peer review of "Synthesis, Characterization and Biological Activities of Zinc Oxide Nanoparticles Derived from Secondary Metabolites of Lentinula edodes"

_molecules, 2023, doi:10.3390/molecules28083532_

Round 1

Reviewer 1 Report

1.      Abstract. The introductory part was complete ignored.

2.      Abstract. The methods shall be clearly explain, the results as well, the results shall be quantitative. Then draw the conclusion in connection with the results

3.      Keywords. The words must be different from title of manuscript

4.      First use the full name then abbreviate throughout the manuscript.

5.      Check the manuscript for typo and grammar errors

6.      The rational of the study is very weak. What is novelty of this study, no explanations, and what type of biological activities are being done to explain the therapeutic targets of ZnO NPs prepared in this study?

7.      The extraction procedure is combination of water and organic solvents, so it is not appropriate to call it the aqueous fraction?

8.      Section 2.2. change the “Effect of pH” with optimization of synthesis of NPs or another heading

9.      The instrumentation including the FTIR, XRD and SEM, the equipment origin was not given

10.    EDS of synthesized NPs were not performed, these studies are mandatory to correlate the results obtained in SEM and also perform the TEM analysis because when the NPs size is more than 10 nm then TEM is mandatory to confirm their sizes

11.    Animal studies, the ethical certificate no and details of ethical committee were not given also how many groups were established to perform the in vivo anti-inflammatory studies,

12.    Antibacterial studies, how many bacterial strains were selected for the studies, no details available, also provide the bacterial strain accession nos.

13.    For various biological studies, IC50 values calculation procedures were not explained

14.     For the analysis by HPLC, no procedure was explained for HPLC, how the analysis was performed

15.    Provide the whole spectrum of ZnO NPs not look like a partial spectrum

16.    Locate the IR peaks for various functional groups present in NPs

17.    Hemolytic activity in the explanation and different while the units are same, confusing, clarify it?

18.    In XRD analysis author should mentioned the hkl value of new generated peaks Also needed to mention the crystal structure and phase composition of prepare NPs.

19.     Why have not the authors carried out a TGA analysis?

20.     Conclusion is just like an end of technical note what was find in present analysis and no future directions were given about the present work 

Author Response

Reviewer 1

Comments and Suggestions for Authors

Respected Sir, I am highly thankful to you for evaluation of my research articles and providing nice suggestion to improve the manuscript, possible suggestion have been resolved

  1. Abstract. The introductory part was complete ignored.

Response: Dear reviewer, thank you for the valuable point. The abstract has been improved according to your suggestion (Line 18-23).

  1. Abstract. The methods shall be clearly explain, the results as well, the results shall be quantitative. Then draw the conclusion in connection with the results.

Response: Respected reviewer, the explanation for methods has been added including results, also connected conclusion with obtained results. (Line 24-26-28-30, 34-39)

  1. Keywords. The words must be different from title of manuscript.

Response: Keywords has been changed according to your suggestion.

  1. First use the full name then abbreviate throughout the manuscript.

Response: Dear reviewer, the change has been made according to your comment. Firstly, added full name then abbreviated throughout the manuscript. (e.g., line 57, 65)

  1. Check the manuscript for typo and grammar errors

Response: The manuscript has been thoroughly revised for English proofreading and grammatical mistakes.

  1. The rational of the study is very weak. What is novelty of this study, no explanations, and what type of biological activities are being done to explain the therapeutic targets of ZnO NPs prepared in this study?

Response: The novelity of study was that secondary metabolites were derived from the novel source Lentinula edodes to manufacture ZnO NPs, no previous study was reported on the synthesis of ZnO NPs from bioactive compound of Lentinula edodes of aqueous fraction. For therapeutic targets of ZnO NPs following biological acvtivities were performed including Antioxidant, Antimicrobial, Anti-inflammatory, Antipyratic, Antidiabetic, Hemolytic.

  1. The extraction procedure is combination of water and organic solvents, so it is not appropriate to call it the aqueous fraction?

Response: In the current study methanolic extract was prepared then methanolic extract was further fractionated by different solvent on the base of polarity (n hexane, chloroform, ethyl acetate, n butanol and water) through separating funnel among these fractions water fraction was used for ZnONPs synthesis.

 As aqueous extraction is extracting components from plant or other material using distilled or deionized water. Organic solvents such as Methanol, Ethanol, Chloroform can also be used for extraction. Fractionation is step by step extraction of components using either of extracting solvents. As we had performed step by step extraction that’s why we called aqueous fraction instead of extract. As aqueous extraction is extracting components from plant or other material using distilled or deionized water. 

  1. Section 2.2. change the “Effect of pH” with optimization of synthesis of NPs or another heading.

Response:  Line 129: Heading change according to your suggestion.

  1. The instrumentation including the FTIR, XRD and SEM, the equipment origin was not given.

Response: Line 140, 147: Equipment’s origin including  FTIR, XRD and SEM has been added.

  1. EDS of synthesized NPs were not performed, these studies are mandatory to correlate the results obtained in SEM and also perform the TEM analysis because when the NPs size is more than 10 nm then TEM is mandatory to confirm their sizes.

Response: EDX results have been added. To measure the size of ZnO nanoparticles we use Zeta sizer which confirm size of nanoparticles. The TEM analysis can not be performed because of the unavailability if facilities. Hence, we would like to request you to allow us to proceed with Zeta sizer as the method for confirmation of size and allow us to not include the TEM analysis. To the best of my knowledge, the purpose to check the size, zeta sizer is also fine.

  1. Animal studies, the ethical certificate no and details of ethical committee were not given also how many groups were established to perform the in vivo anti-inflammatory studies,

Response: (Line 537-541) For in vivo anti-inflammatory activity 3 groups were selected including Diclofenac

Sodium Standard as positive control one is Negative control and third one is for ZnO nanoparticles. Furthermore, the ethical approval numbers has been provided at the respective section “ethical approval”.

  1. Antibacterial studies, how many bacterial strains were selected for the studies, no details available, also provide the bacterial strain accession nos.

Response: Line 197-198: Bacterial strain accession numbers has been added. The chosen bacterial species includes Gram-positive (Staphylococcus. aureus) and Gram-negative bacteria (E. coli) and Klebsilla. pneumoniae.

  1. For various biological studies, IC50 values calculation procedures were not explained

Response: Line 162-163: Procedure of IC50 has been added in section 2.5.1.

  1. For the analysis by HPLC, no procedure was explained for HPLC, how the analysis was performed.

Response: Line 104-111: HPLC analysis procedure has been added.

  1. Provide the whole spectrum of ZnO NPs not look like a partial spectrum.

Response:  Figure 1: Complete spectrum has been added of ZnO NPs.

  1. Locate the IR peaks for various functional groups present in NPs

Response: Figure 2: IR peaks of various functional groups has been shown.

  1. Hemolytic activity in the explanation and different while the units are same, confusing, clarify it?

Response: Line 345-351: The query has been resolved.

  1. In XRD analysis author should mentioned the hkl value of new generated peaks Also needed to mention the crystal structure and phase composition of prepare NPs.

Response: Line 276-286: Hkl values has been added in XRD spectra. Composition and crystalline structure also mention in results.

  1. Why have not the authors carried out a TGA analysis?

Response: As TGA use to check thermal stability and we use green route in which bioactive compounds involve for synthesis of ZnO NPs confirm through HPLC and FTIR. As the upper temperature used for TGA is normally 1000°C at this temperature bioactive compounds will degraded

  1. Conclusion is just like an end of technical note what was find in present analysis and no future directions were given about the present work 

Response: Line 502-516: Conclusion has been modified according to your suggestion.

Reviewer 2 Report

Manuscript entitled " Synthesis, characterization and biological activities of Zinc....... Lentinula erodes" by Amin is interesting subject but repetitive concept. Requires manor revision in the following points

1) Make figure 1 compact.

2) Why FTIR spectra did not show the active curves.

3) Figure 3 control data is missing. Please show the zinc alone curve.

4) SEM image is not clear. Show the individual particles rather than mass.

5) Discussions portion is loosely connected with the obtained results. Requires further modifications.

Author Response

Reviewer 2

Comments and Suggestions for Authors

Manuscript entitled " Synthesis, characterization and biological activities of Zinc....... Lentinula erodes" by Amin is interesting subject but repetitive concept. Requires manor revision in the following points

Response: Respected Reviewer, I am highly thankful to you for evaluation of current research article and providing nice suggestion to improve the manuscript, possible suggestion have been addressed.

1) Make figure 1 compact.

Response: Respected reviewer, new figure has been added according to your suggestion.

2) Why FTIR spectra did not show the active curves.

Response: HPLC data of aqueous extract represents the presence of phenolics compounds which may responsible for the ZnO NPs while FTIR spectrum of ZnO NPs shown the peaks of OH and COOH functional groups at given wavelength, these functional groups supports the compounds detected in HPLC analysis while OH functional group showm broad peak and carboxylic acid have sharp  peak.

3) Figure 3 control data is missing. Please show the zinc alone curve.

Response: The figure 3 has been improved.

4) SEM image is not clear. Show the individual particles rather than mass.

Response: New images of SEM has been added with fine spectra of ZnO NPs.

5) Discussions portion is loosely connected with the obtained results. Requires further modifications.

Response: Line 426, 435-440, 444-445, 461-498: Discussion has been improved with obtained results.

Round 2

Reviewer 1 Report

Manuscript in its current form reached the level for final acceptance for publication

Reviewer 2 Report

All my comments are addressed by authors, hence manuscript can be accpted.